

# Modeling ground deformation associated with the destructive earthquakes occurring on Mt. Etna's southeastern flank in 1984

Flavio Cannavò[1], Salvatore Gambino[1], Biagio Puglisi[2], Rosanna Velardita[2]

[1]*Istituto Nazionale di Geofisica e Vulcanologia, - Osservatorio Etneo, Piazza Roma 2, Catania, Italy*

[2]*Istituto Nazionale di Geofisica e Vulcanologia, - Osservatorio Etneo, Via Monti Rossi, 95123 Nicolosi (CT), Italy*

*Submitted to Natural Hazards and Earth System Sciences*

**Correspondence to:**

Dr. Salvatore Gambino
Istituto Nazionale di Geofisica e Vulcanologia
Sezione di Catania Osservatorio Etneo
P.zza Roma 2, 95123 Catania, Italy
tel. number 39-95-7165877
fax number: 39-957165826
e-mail: Salvatore.gambino@ingv.it







**Abstract**
The Timpe Fault System is the source of very shallow but destructive earthquakes that affect
several towns and villages on the eastern flank of Mt. Etna (Italy). In 1984, several seismic events,
and specifically on October 25, caused one fatality, 12 injuries and produced serious damage in the
Zafferana and Acireale territories. This seismicity was mainly related to the activity of the Fiandaca
Fault, one of the structures belonging to the Timpe Fault System.
We inverted ground deformation data collected by a geodimeter trilateration network set up in 1977
at a low altitude along the eastern side of the volcano in order to define the Timpe Fault System
faulting mechanisms linked to the seismicity in 1984.
We found that in the May 1980-October 1984 period, the Fiandaca Fault was affected by a strike
slip and normal dip slip of about 27 and 23cm. This result is in fairly good accord with field
observations of the co-seismic ground ruptures along the fault but it's notably large compared to
displacements estimated by seismicity, then suggesting that most of the slip over the fault plane was
aseismic.
The results once again confirm how seismicity and in particular ground ruptures represent a very
high hazard to the several towns and villages situated along the Fiandaca Fault.
***Keywords:*** *Geodetic observations, flank dynamics, fault displacements, shallow earthquakes*

**1.0 Introduction**
The determination of finite fault geometry and slip for severe earthquakes is important for the
mitigation of seismic hazard and in particular for very shallow earthquakes that entail surface
fracturing.
On the eastern flank of Etna volcano, movements along very shallow normal faults generate
recurrent seismicity sometimes leading to destructive events, since densely urbanized areas are



located on these structures (e.g. Azzaro et al., 2012; Barreca et al., 2013c). The Timpe Fault System
(TFS) is the main source generating the strongest earthquakes known over the last 200 years
(Azzaro et al., 2012).
The TFS dissects the southern-eastern flank of Mt. Etna (Fig. 1) and is formed by several fault
segments that include the Fiandaca (FF), S. Tecla (STF), S. Venerina (SVF), Moscarello (MF) and
San Leonardello (SLF) faults (fig. 2B), generally characterized by normal and right-lateral
dynamics (Azzaro, 1999; Azzaro et al., 2000).
The TFS shows right-lateral and normal dip-slip kinematics and each fault segment is characterized
by earthquake related displacements of tens of centimeters and aseismic sliding behavior with slip
rates of few mm/year (Bonforte et al., 2011; Azzaro, 2004).
Shallow seismicity (< 3 km), associated with these faults, includes the occurrence of several
earthquakes with magnitude up to 4.5 (Azzaro et al., 2000) with epicenter in the fault areas. Despite
their low-energy release, these events produced destructive effects, with fatalities and injuries, in a
very narrow area where they can reach epicentral macroseismic intensity $I_0$ up to VIII/IX EMS
(European Macroseismic Scale, Grünthal, 1998), often accompanied by coseismic surface
fracturing (e.g. Azzaro, 1999; 2004) with a mean recurrence time of about 20 years (Azzaro et al.,
2013c). The last destructive episodes with $I_0$=VIII were recorded on 25 October 1984 and 29
October 2002. The first is the subject of this work while the second, linked to the SVF dynamics,
caused damage at Santa Venerina village (Azzaro et al., 2006; La Delfa et al., 2007).
During the second half of October 1984, an anomalous large number of seismic events,
concentrated mostly on the eastern side of the volcano, were recorded on Mt. Etna (Gresta et al.,
1987). The main events occurred in particular on 19 and on 25 October 1984, respectively at 17.43
and 01:11, with epicentral macroseismic intensity $I_0$ VII (19.10) and VIII (25.10) EMS, which
struck the territory of Zafferana and in particular the villages of Fleri and Pisano.
The 25 October earthquake was linked to the activity of the Fiandaca Fault (FF) (Azzaro, 1999),
while that on 19 October was located in proximity of the Santa Tecla fault (STF). In this paper, we



examine the slope distance measurements collected between 1977 and 1985 by a geodimeter
trilateration network (Ionica Network, Fig, 2A) set up in 1977 along the eastern edge of the volcano
at low altitude (0-700 m b.s.l.) and measured until 1985, to shed new light on the kinematic aspects
of this sequence of earthquakes. Indeed, the previously unpublished data, except for an internal
report (AA.VV., 1985 in Italian), have been reviewed in the wake of new knowledge acquired in the
last two decades, enabling insights into Etna's eastern flank dynamics that were not possible at the
acquisition time.

**2.0 Mt. Etna and its structural framework**
Mount Etna (Fig. 1) is a large basaltic volcano built up in a geodynamic setting generated during the
Neogene convergence between the African and European plates (e.g. Allard et al., 2006; Branca et
al., 2011). It is situated on the eastern coast of Sicily and is one of the most active volcanoes in the
world. Mount Etna's activity may be grouped into two types: persistent activity comprising
degassing phases alternating with strombolian activity, which may evolve into lava fountains and
effusive events, and lateral flank eruptions occurring along fracture systems that are generally
preceded by an intrusive process (e.g. Aloisi et al., 2006). The volcano is located at the intersection
of two main regional fault systems, having NNW–SSE and NE–SW trends respectively (Fig. 1).
The NNW–SSE structural system represents the Sicilian onshore continuation of the Malta
Escarpment (ME), the major crustal-scale fault separating the continental African platform from the
oceanic Ionian Basin. Ripe Della Naca in Fig. 1 represents surface evidence related to NE-SW
Messina-Fiumefreddo (MF) line, while the faults of the "Timpe" system are the major tectonic
surface manifestations of the ME. The interaction between regional stress, dike-induced rifting and
gravity force is the cause of a fairly continuous and roughly eastward and downward motion of its
eastern flank (e.g. Puglisi and Bonforte, 2003; Solaro et al., 2010). This sliding area (Fig. 1) is
delimited to the north by the Pernicana–Provenzana Fault System (PFS) (e.g. Obrizzo, 2001;




Alparone et al., 2013a), a transtensive E–W trending complex active tectonic structure, while the
Trecastagni and Tremestieri faults (e.g. Bonforte et al., 2013) represent the main southern
boundaries. The Mt. Etna GPS network has enabled us to determine how the entire eastern flank is
affected by an ESE-ward motion, at a mean rate of about 1-3 cm/year (e.g. Bonforte et al., 2011;
Gambino et al., 2011). Moreover, starting from 1980, the sliding area underwent two marked
acceleration phases in October 1984–1986 and in October 2002 -2005, as described in Alparone et
al. (2013c). The authors also highlight a significant temporal correlation between periods of flank
acceleration and intensified seismic activity. The TFS is very active from the seismic point of view,
both for the number of events and for maximum intensity reached at the epicentre (Azzaro et al.,
2013a). TFS is characterized by surface faults of considerable length (up to 8–10 km) and scarps
(Azzaro, 1999; Lanzafame et al., 1994). It includes (Fig. 2B) N140°E striking faults (FF, STF and
SVF), which define normal right-lateral structures extending from the town of Acireale to Zafferana
Etnea and the MF and SLF faults with N160°E structural trend that dissect the base of the volcano's
flank by prevailingly vertical movements (Azzaro et al., 2013a).
The TFS plays a key role in the local tectonics accommodating the ESE motion; this fault system
divides the sliding sector into several blocks (Solaro et al., 2010; Bonforte et al., 2011)
characterized by homogeneous kinematics with relative motion measured by Permanent Scatterers
(PS) along the faults of about 3-5 mm/years in a "quiet" period such as 1995-2000 (Bonforte et al.,

122  2011).


**3.0 Mt. Etna seismicity during 1984**
In October 1984, an intense seismic sequence was recorded on Mt. Etna area that marked an
unusual behavior of the volcano. This swarm comprised more than 1000 earthquakes with M > 2.0
over two weeks (16-30 October) and that involved the summit area and almost the entire eastern
flank (Gresta et al., 1987; Gresta and Musumeci, 1997).



In particular, thousands of events, largely concentrated on the eastern side of the volcano, occurred
from October 19 to October 31 (Gresta et al., 1987). The main events (Fig. 2B) were on 19[th] (at
17:43) and 25[th] October 1984 (at 01:12), and struck the town of Zafferana and the Fleri and Pisano
villages (Fig. 3).
The earthquakes caused one fatality in Zafferana and injured twelve people in Fleri. In Fleri, the
number of injuries would have been much higher, had most of the inhabitants, in great anxiety after
an initial shock at around 22.00 the day before, not spent the night outdoors. Serious damage was
caused to buildings in Zafferana but particularly in Fleri, which was almost entirely destroyed.
About 70% of the buildings of the entire municipality (including all public buildings) were declared
unfit for use (Fig. 3).
Figure 2 reports the location and epicentral macroseismic intensity ($I_0$) as reported in the
Macroseismic Catalogue of Mt. Etna (CMTE Working Group, 2008). An $I_0$ of VII EMS (European
Macroseismic Scale, Grünthal, 1998) is reported for the October 19[th] event and of VIII EMS for the
October 25[th] event.
In October 1984 more than 1.5 km long NW-SE trending cracks extended from the village of Fleri
with dip-slip displacements of about 20 cm. It is worth noting that a similar ground rupture affected
the southeastern part of FF on occasion of the VIII EMS event of June 19 1984 (Azzaro, 1999).
The volcanic district of Mt. Etna, and in particular its eastern flank, is affected by earthquakes
characterized by a strong attenuation of seismic energy in an orthogonal direction to the fault plane.
This produces damage extending along narrow zones (1-5 km long, up to 1 km wide) around the
seismogenic source (Azzaro et al., 2006) and coseismic surface faulting effects, reported in detail in
historical accounts, for the major seismic events. Regarding FF, earthquakes occurring in 1875,
1894, 1907, 1914 and 1931 (Azzaro, 2004) caused NNW-SSE trending fractures that opened for
several kilometers with prevailing extensional movement and right-lateral displacements of several
centimeters.



Figure 2B reports macroseismic and instrumental locations of the main seismic events between May
1980 and October 1984 in the TFS area. Given that the macroseismic epicentre is calculated as the
barycentre of the data points with intensity I = $I_0$, $I_0$-1, the macroseismic location and the
instrumental location may be rather different, (Azzaro et al., 2000). Indeed Gresta et al. (1987),
using a seismic network of just a few active stations in 1984, estimated a duration magnitude of 4.2
and 3.9 for the Zafferana and Fleri earthquakes respectively with different epicentres (Fig. 2B).

**4.0 The "Ionica" EDM network**

Between 1977 and 2002, the monitoring of the horizontal component of ground deformation at Mt.
Etna was carried out by trilateration geodetic techniques using EDM (electro-optical distance
measurements). Three separate networks were installed on the northeastern, western, and southern
flanks (Fig. 1).
A fourth EDM network (Ionica Network), was installed and measured for the first time in October
1977 (Fig. 1), with the aim of verifying the possible relationships between the regional tectonic
activity, highly evident in this area with the presence of numerous structures, and the volcanic
activity (Fig. 2B).
The Ionica Network was set up along the eastern edge of the volcano, along a line from Catania to
Taormina, at low altitude (between 700 and sea level); it consisted of 19 benchmarks and 43 lines,
which were measured yearly from 1977 to May 1980.
The slope distances were recorded by using an AGA 6BL Laser Geodimeter (Fig. 1), and were
corrected for atmospheric conditions considering temperature and atmospheric pressure values
acquired at the measurement points. The instrumental error of such measurements is 5 mm plus 1
ppm of the surveyed distance.
After the Zafferana and Fleri 1984 earthquakes, four measurement surveys were performed in the
period October 1984 - March 1985 in the portion of the network (Fig. 2B) covering the area
affected by the events. The first survey was made between 26 and 31 October. These measurements





involved only the southern part of the network, consisting of 9 benchmarks and 19 lines (Fig. 2B)
whose mean length is 4.7 km.

**4.1 EDM Data**
The results obtained from data collected from 1977 until 1985, indicate that the variations of
distance with values greater than instrumental error occurred mainly in the period May 1980-
October 1984, while distance variations obtained from the comparison with the other surveys are
mainly within the error. In particular, changes are up to 108 mm and ten measurements showed
variations over 50 mm (Fig. 4). In the previous period (1977-1980), the changes observed are
generally within the instrumental error though a trend is detectable for several lines.
Ground deformation strain field is given by the uniform strain tensor components $\varepsilon_{ij}$ which can be
calculated by using variations of slope distances (Jaeger, 1969):
$\Delta L_{MN}/L_{MN} = \varepsilon_{11} \cos^2 \delta_{MN} + \varepsilon_{22} \sin^2 \delta_{MN} + \varepsilon_{12} \sin^2 \delta_{MN}$
where $\Delta L_{MN}$ is the change in length of the line MN (with length $L_{MN}$) between two points M and N
and $\delta_{MN}$ is the angle between MN direction and x-axis.
This tensor indicates the average deformation occurring between two different surveys in the area
covered by the network and provides useful information on the ground deformation regime of the
area (e.g. Bonaccorso, 2002). We calculated the principal strain axes (Fig. 2C) drawn from the
comparison of the overall measurement interval 1980-October 1984. We obtained a positive
extension ($\varepsilon_1 = 17 \pm 4.4$ μstrain) oriented approximately orthogonal to and a contraction ($\varepsilon_2 = -12.2$
$\pm 4.4$ μstrain) parallel to the FF and STF fault systems (Fig. 2).

**5.0 Geodetic Data Modelling**





The surface ground deformation field for the 1980-1984 interval was used as input to
constrain isotropic half-space elastic inversion models using Okada's (1992) model. The aim of this
inversion is to characterize the FF and STF dynamics during this particular period. Unlike in 1985,
several of the geometric parameters of the two considered faults are known today (Azzaro et al.,
2013). Hence, reducing the unknowns, enables one to make an inversion of the limited EDM
dataset.
We fixed the more reliable parameters (Tab. 1), while for the more dubious ones (less
precise), we chose to leave them to be free (in a range) during the inversion together with the
kinematic parameters.
To model the displacements due to each single fault, we adopted the analytical model
described in Okada (1992), and to take the simultaneously effects of two (or more) faults into
account we used the superposition principle. The Okada equations give the 3D displacement ($\delta_P$) at
a point P due to the fault geometry and its kinematic. For a baseline between the points Pi and Pj,
we modelled the EDM distance variation as:

$\Delta_{ij} = \left\| \left( P_i + \delta_{P_i} \right) - \left( P_j + \delta_{P_j} \right) \right\| - \left\| P_i - P_j \right\|,$
where $\|\cdot\|$ is the 3-dimensional Euclidean norm operator.

In particular, we inverted for the parameters reported in gray in Table 1, where the
associated ranges are also shown. These values are estimations, calculated from geophysical,
geological and historical data; however, not all parameters are available. We excluded all the lines
crossing the MF. Thus we had a total of 10 free parameters and 13 EDM measurements. It is worth
noting in Table 1 that 5 free parameters reach an extreme in their feasible ranges. Hence, they could
be treated as fixed parameters and set to their maximum/minimum possible values. We considered a
shear modulus of 10 GPa and a Poisson's ratio of 0.25 (e.g. Bozzano et al., 2013).



In order to estimate simultaneously the free geometric parameters and kinematics of both the
considered faults, we inverted the EDM data by minimizing the weighted misfit between the
measured and calculated distances. Because of the free geometric parameters, the mathematical
problem is nonlinear and the adopted minimization algorithm was a hybrid approach of genetic
algorithm and pattern search (Audet & Dennis, 2002). The measurements were weighted with their
associated instrumental errors expressed in meters by the formula:
$\sigma_{\Delta_{ij}} = \sqrt{2}\big(0.005 + 10^{-6}\|P_i - P_j\|\big).$
The square root of 2 is due to the error propagation in calculating the distance variation $\Delta_{ij}$
neglecting the displacement.
The found model fits the EDM data with a WRMS of 0.98.
A Jackknife re-sampling method (Efron, 1982) was used to estimate the error of model
parameters. The technique requires several optimization executions, each one deprived of just one
measurement in input. The errors at 99% are estimated as 3-times the standard deviation in the set
of the found solutions.
The final results (Tab. 1 and Fig. 5) are in agreement with the dominant faulting style
producing a normal strike-slip movement with 27.6 cm of dextral strike movement and 22.7 cm of
normal dip on FF, while we obtained only 6.0 cm of dextral strike slip on STF. A comparison
between observed and modeled slope distance is reported in Fig. 6.

**5.1. Sensitivity analysis**
We carried out a sensitivity analysis in order to ascertain whether our data could constrain a
valid set of fault parameters. We adopted the Sobol' analysis (Sobol', 1990), a variance-based
global method to measure sensitivity across the whole input space, deal with nonlinear responses,
and estimate the effect of interactions in non-additive systems. The method breaks the variance of
the output of the model down into fractions (normalized to 1) which can be attributed (in



percentage) to input terms. The Sobol' first-order indices indicate the contribution of the main
effect of each input parameter to the output variance, therefore measuring the effect of varying the
input parameter alone, averaged over variations in other input parameters. We adopted the
algorithm in Cannavò (2012) to calculate the first-order Sobol' indices of all the fault parameters
given in our EDM data. The indices reported in Table 2 represent the fraction of variance in the data
that can be attributed to each fault parameter. The higher the fraction, the more constrained is the
parameter by the data. Results show that, among all the parameters, the data are more sensitive to
fault lengths and mainly to strike-slips which, in turn, can be estimated more accurately than the
other parameters.

**6.0 Discussion and conclusions**
The Timpe is a normal fault system dissecting the Mt. Etna's lower eastern flank. It is
formed of several segments (FF, STF, SVF, SLF and MF) that show right-lateral kinematics and
normal dip-slip with slip-rates ranging from 3.0 to 5.0 mm/y (Bonforte et al., 2011).
Timpe dynamics are linked to the ESE-ward motion of the eastern flank of Mt. Etna (e.g.
Azzaro, 2013a). Since 1980, the ESE-ward motion has shown phases of increased velocity, the first
being observed in the October 1984-1987 period (Alparone et al., 2013c). The TFS accommodates
this motion and becomes very active from the seismic point of view when acceleration phases
characterize the ESE-ward motion of the eastern flank.
These severe/destructive events, with a mean recurrence time of about 20 years for Azzaro
et al. (2013c), make the Timpe fault system extremely important in terms of seismic hazard.
These events are distributed on the several segments and in the last decades the destructive
episodes have affected FF in October 1984 and SVF in November 2002.
The 25th October 1984 event most likely represents the strongest event recorded on FF since
1875 (Azzaro et al., 2004); we tried to characterize the finite fault geometry and slip of FF by



ground deformation observations. We recovered and analyzed the EDM measurements of the
"Ionica" network crossing the TFS and measured only between 1977 and 1985. These data
highlight major variations between 1980 and 1984; the principal components of the strain tensor
obtained in this time period, show a positive extension oriented approximately orthogonal and a
negative extension parallel to the FF and STF fault systems and are consistent with normal right-
lateral dynamics of the two structures.
We inverted data showing that, between 1980 and November 1984, the FF (7 km length and
2.6 km depth) was affected by a strike slip component of 26.7±1.5 cm and a normal dip slip of
22.7±2.4 cm (overall displacement ca. 35 cm). The sensitivity analysis indicates that fault length
and strike-slip represent the most accurately constrained parameter obtained by data inversion.
In consideration of these results, and assuming a medium rigidity ($\mu$) of a shear modulus of
10 GPa in the general relation (Aki, 1966):

$Mo = \mu * S * \bar{u}$
we obtained a geodetic moment $M_G = 6.0 * 10^{23}$ dyne-cm.
An estimate of the seismic moment release associated with the seismic events was obtained
using the relation (Giampiccolo et al., 2007) for Etnean earthquakes:

$Log(Mo) = (17.60 \pm 0.37) + (1.12 \pm 0.10) * M_L$
It shows that Mo cannot be greater than $= 1.6 * 10^{23}$ dyne-cm for the 25 October 1984
earthquake and $4.0 * 10^{22}$ dyne-cm for that on 19 June 1984. Therefore only a part (from 5% to a
maximum of 30%) of the stick-slip obtained by modeling is related to the co-seismic effects of the
earthquakes recorded along the FF, suggesting that most of the slip over the fault must be aseismic.
These results are similar to those obtained by Obrizzo et al., (2001) for the Pernicana Fault System,
on analyzing 17 years of levelling data. These authors concluded that only 30% of the total
deformation of Pernicana Fault System is attributable to co-seismic displacements.



Data inversion cover a long period (1980-84); however from May 1980 to May 1984, no events
with epicentral intensities $I_0 \geq VI$ EMS were recorded (Fig. 2B) along FF that probably was only
subjected to a few centimeters of aseismic displacement (taking in consideration the 3-5 mm/year
measured by PS by Bonforte et al., (2011). An acceleration of the FF dynamics could have occurred
since June 1984 when an EMS VII was recorded on southeastern part of the fault (Fig. 2B) or after
October, 16 when more than 1000 earthquakes with M > 2.0 took place on the eastern flank in two
weeks. Field observations certainly suggest that co-seismic ground displacements are larger than
what would be expected from the June 19 and October 25 earthquakes. If we exclude any coseismic
effect, it is difficult to determine when the dynamics of the FF underwent an acceleration in June-
October 1984. If this were the case, it could mean that a destructive seismic event, such as the 25
October 1984 one, might have been preceded by a measurable increase of the ground deformation
close to the fault.
Few variations have been detected on the Santa Tecla Fault (STF), which seems to have been little
affected by the October 1984 dynamics. The June 19 event does actually appear to be located at the
northern edge of the fault (Gresta et al., 1987). The model estimated only 6.0 (±2.0) cm of strike
slip on STF. Then the October 19 event does not seem related to significant shifts of the STF.
Precise locations of the earthquakes ($M_{max}=3.7$) recorded in the same area between 1995 and 2006
(Alparone and Gambino, 2003; Alparone et al., 2013b) have shown that the seismicity in this sector
is generally 3-5 km b.s.l. deep and related to NE-SW oriented seismogenetic structures (MF line).
For all these reasons we retain that the October 19 event is not attributable to TDF dynamics.
To summarize, the destructive 25 October 1984 event is an effect of an important dynamic episode
that affected the FF structure time related with a flank acceleration phase (Alparone et al., 2013c).
EDM data inversion indicated a total stick-slip of ca. 30-35 cm between 1980 and 1984, mainly
between June and October 1984 along a fault length of 7 km. This result is in some ways
comparable with field observations that detected a co-seismic ground rupture of the northwestern





and southeastern sectors of the fault of up to 20 cm (Azzaro, 1999 and reference therein), while a
discrepancy between the seismic and geodetic moment is present. Indeed, FF shows a low seismic
efficiency (lower than 30%), a feature that seems common to other very shallow faults on the
eastern flank (e.g. the Pernicana Fault System) and that highlights that most of the energy is
involved in aseismic ground displacements.
These considerations again confirm the high level of seismic risk, in particular ground rupture
hazard of the Fiandaca Fault and generally of the Timpe Fault System, for the several towns and
villages located on these structures.
**Acknowledgements**
We are indebted to Prof. Letterio Villari who, since the 1970s, understood the importance of
monitoring Etna's eastern flank, and who planned and worked to set up the "Ionica" EDM network.

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



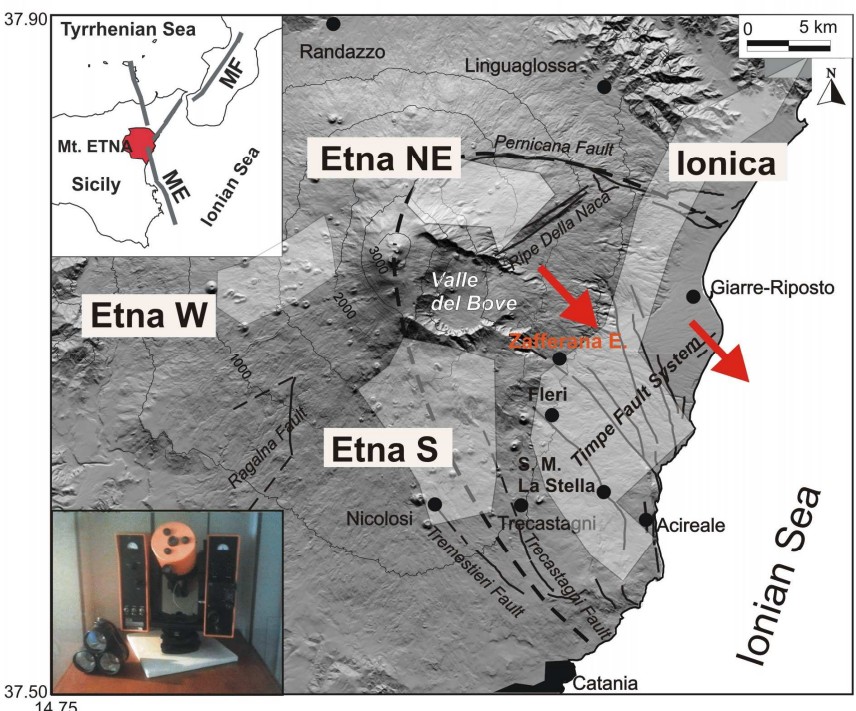


**Fig. 1** –Surface faults map of Mt. Etna. Top inset map shows the main regional fault

systems: MF=Messina-Fiumefreddo line, ME=Malta Escarpment. Dashed line defines the

sliding sector and red arrows indicate its movement direction. White areas are covered by EDM

networks. In the bottom insert, the AGA 6BL Laser Geodimeter.





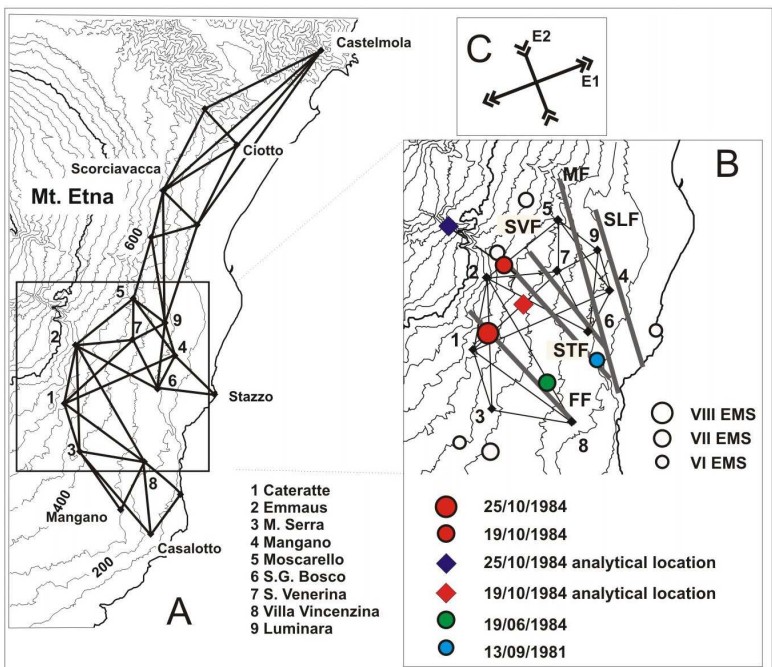

**Fig. 2.** – Ionica EDM network (a) EDM benchmarks and lines measured between 1977 and
1985 (b) Macroseismic epicentres of earthquakes with epicentral intensities $I_0 \geq$ VI EMS occurring
from 1980 to October 1984 in the south-eastern flank of Mt Etna. (c) Principal strain axes obtained
from the comparison of the overall measurement interval 1980-October 1984.

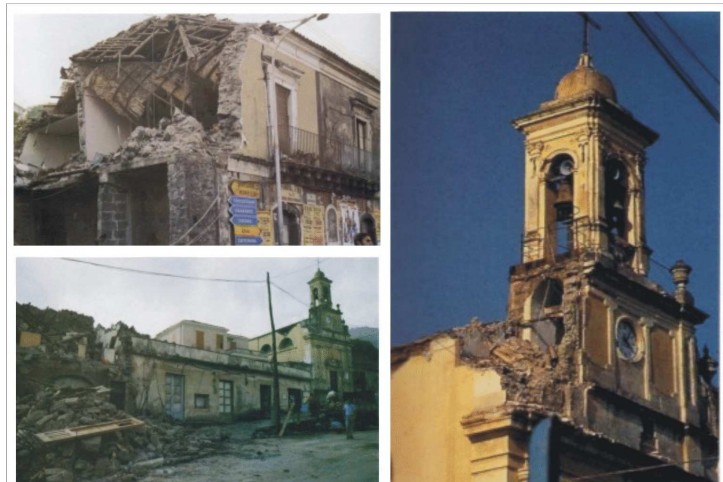


**Fig. 3.** Photos of damage caused during the 25[th] October earthquake at Fleri village
(http://www.ct.ingv.it/macro/etna/html_index.php).




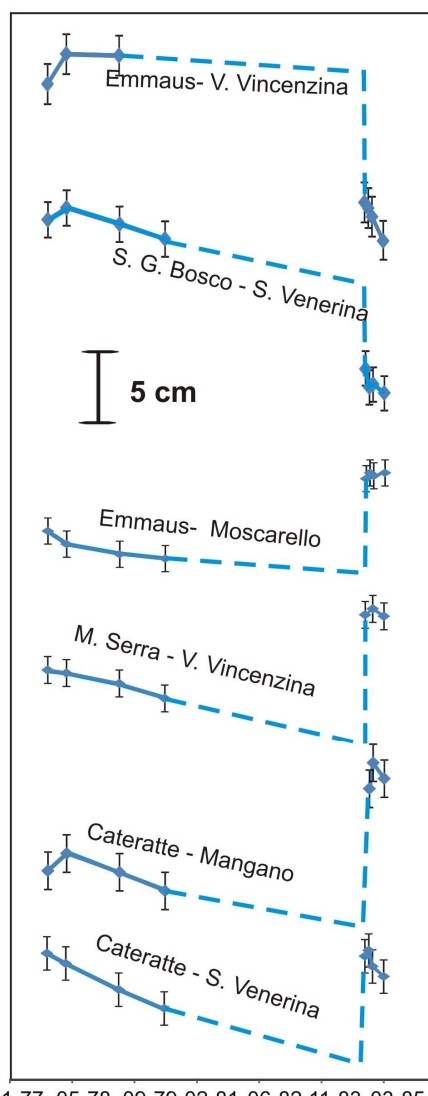

454  01-77 05-78 09-79 02-81 06-82 11-83 03-85

**Fig. 4.** Changes of measured line lengths with respect to time of surveying time (interval
1977– 1985) in the "Ionica" network. Note how the lines are subject to marked variations in the
1980-1984 period. Solid lines connect measurements and dashed lines represent the plausible trend
of the May 1980-October 1984 period.





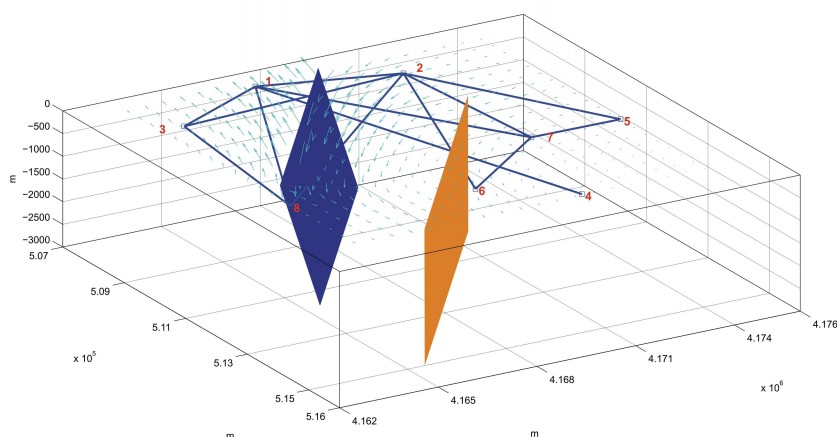


**Fig. 5.** Location and E-W cross-section of the source modelled. The arrows represent the
simulated deformation field due to the estimated fault kinematics. Numbers as in Fig. 2.

**Fig. 6.** Comparison between observed (blue bars) and modeled (red bars) slope distance.
Numbers as in figure 2.









**Table 1.** Best Fitting Range for the model.

|  | Fault 1 (FF) | | Fault 2 (STF) | |
|---|---|---|---|---|
| X (m, center top) | 509700 | fixed | 512423 | fixed |
| Y (m, center top) | 4166660 | fixed | 4168595 | fixed |
| Depth (m, top) | 0 | fixed | 0 | fixed |
| Azimuth (°) | 140 | fixed | 140 | fixed |
| Dip (°) | 70 ± 0.0 | 70 ÷ 89.9 | 89.9 ± 25 | 70 ÷ 89.9 |
| Semi-Length (m) | 3500 ± 0.0 | 2000 ÷ 3500 | 4000 ± 0.0 | 2000 ÷ 4000 |
| Width (m) | 2610 ± 70 | 1000 ÷ 3000 | 3000 ± 100 | 1000 ÷ 3000 |
| Strike-s (cm) | 26.7 ± 1.5 | 0 ÷ 100 (dextral) | 6.0 ± 2.0 | 0 ÷ 100 (dextral) |
| Dip Slip (cm) | -22.7 ± 2.4 | -100 ÷ 0 (normal) | 0.0 ± 3.0 | -100 ÷ 0 (normal) |


**Table 2.** Global sensitivity analysis. First-order Sobol' coefficients for the fault parameters

and total sums**.**

|  | Fault 1 (FF) | Fault 2 (STF) | TOTAL |
|---|---|---|---|
| Azimuth (°) | 0.000 | 0.002 | 0.002 |
| Dip (°) | 0.000 | 0.005 | 0.005 |
| Semi-Length (m) | 0.119 | 0.126 | 0.245 |
| Width (m) | 0.012 | 0.019 | 0.031 |
| Strike-s (m) | 0.125 | 0.284 | 0.409 |
| Dip Slip (m) | 0.013 | 0.014 | 0.027 |
| TOTAL | 0.269 | 0.450 |  |
