# Peer review of "Modeling ground deformation associated with the destructive earthquakes occurring on Mt. Etna's southeastern flank in 1984"

_Natural Hazards and Earth System Sciences, 2015_

## Referee Comment (RC1) · M. Bonafede (Referee) · 10 Feb 2016

Please see the attached pdf file for displayed equations

Comments from Maurizio Bonafede – Department of Physics and Astronomy, University of Bologna, Italy

Introduction

The Timpe fault system (TFS), is thought to accommodate (1) volcanic inflation in the East flank of Mt Etna, due to magma input/output and (2) the motion of the major tectonic lineaments in the region: the Messina-Fiumefreddo and the Malta escarpment (with its Northern prosecution as the Tindari-Giardini lineament). Of course both

mechanisms (1) and (2) must be considered as interacting processes, and it is difficult to assess what proportion of either mechanisms is responsible for the observed slip of the TFS. The problem is particularly awkward due to

(i) the scarcity of geodetic data, acquired yearly in a limited number of campaigns from 1977 to 1980 and again only after the earthquakes in 1984, considerd in this study;

(ii) the presence of several recognized faults in the area (5 are mentioned as subfaults of the TFS, but others are present within a slightly larger domain, as shown in figure 1);

(iii) the mentioned role of volcanic inflation/deflation, often with short time scales, which superpose large transient signals to the presumably more regular tectonic motions.

(iv) the scarcity of coherent points in the Eastern flank, where SAR data (in the years following 1990) might (in principle) constrain a more uniformly distributed deformation field.

For instance, Trasatti et al. (2009, Geophys. J. Int., 177, 806-814 – doi: 10.1111/j.1365-246X.2009.04093.x) describe the deformation of Mt Etna during 1993-1997 in terms of volcanic inflation and a rigid body translation of a sliding sector of the eastern flank, between the Pernicana fault (N of TFS) and the TFS. They employed GPS, tilt an SAR data but did not attempt to split the E-sector in differently sliding subsectors because of the increasing number of unknown parameters in the inversion.

Main comments

(A) Several earthquakes struck the E flank of Mt Etna between 1980 and October 1984 (only one is mentioned in June 19-th, 1984 but it is not accounted for); these may add transient components unresolved in the geodetic data between 1980 and 1984). Furthermore, as said already, deformation in a volcanic environment cannot be assumed as a steady state process, being related to episodic inflation/deflation episodes. For this reason, the straight dashed lines shown in figure 4 are meaningless.

(B) The formula at line 192 is wrong (hopefully it is only a misprint, otherwise the inversion procedure should be re-executed): the last term should read $\varepsilon\_12 \sin2\delta$ (not $\varepsilon\_12 \sin\hat{~}2\ \delta$). It must be stated clearly that the deformation computed in this way is the "equivalent" uniform deformation (as reported in figure 2.C) providing the same distance variations as the real non-uniform deformation, concentrated on the faults

(C) Why are data at benchmarks 4 and 9 not taken into account? The number of free parameters (10) is so close to the number of independent data (13) that it is difficult to assess the reliability of the inversion. The more so, since data were arbitrarily (if I understand correctly, according to statement A above) corrected assuming a steady-state creep (dashed lines in figure 4) which is not supported by real data. Furthermore, some fault parameters (fault depth, length, dip) are fixed a priori. The data clearly show post-seismic creep and a major creep event is mentioned before the earthquakes (page 6). What would be the result of the inversion if the real data (1984 minus 1980) were considered?

Minor points

1. line 44: better write "... along the Timpe Fault System" instead of "... along the Fiandaca fault".

2. Lines 83-84: the previously unpublished data ... have been reviewed in the wake of new knowledge acquired in the last two decades (explain: what new knowledge? is it the fault parameters mentioned al line 206?), enabling insights into Etna's eastern flank ...

3. The magnitude of the seismic events should be always stated when they are first mentioned.

4. line 297: I do not get the mentioned Mo values employing the magnitudes m=4.2 and m=3.9 mentioned in the text. Furthermore, these are duration magnitudes, not Richter magnitudes mL. In any case, it is clear that most of the fault displacement is

aseismic.

5. line 308: EMS VII is written here for the June '84 event while it is rated as VIII at page 6.

6. explain acronym TDF at line 323; I cannot find it elsewhere;

7. the acronym MF is employed for both the Messina-Fiumefreddo line and for Mascarello fault: consider revising.

8. Figure 2: consider reporting in the caption the acronyms of the faults. Eliminate topographic level lines from panel B. Write "instrumental epicenter" (instead of "analytic location") and "macroseismic epicenter" otherwise (if I understand correctly).

9. Figure 3 is unnecessary: consider deleting, leaving the web link in the text.

Please also note the supplement to this comment:
http://www.nat-hazards-earth-syst-sci-discuss.net/nhess-2015-312/nhess-2015-312-RC1-supplement.pdf
* * *

---

## Referee Comment (RC2) · Anonymous Referee #2 · 16 Feb 2016

The Authors use a geodetic dataset based on ground deformation data collected by a geodimeter trilateration network for assessing the co-seismic and aseismic slip along one of the most active fault of Mt.Etna, in a period including a strong earthquake. I think the topics could be suitable for the journal after revisions providing some elements of broader interest for the readers. For example it is necessary to frame the aseismic slip in process of gravitational sliding involvig the eastern flank of Mt. Etna. In particular , the Authors assert that "in the May 1980-October 1984 period, the Fiandaca Fault was affected by a strike slip and normal dip slip of about 27 and 23cm. This result is in fairly good accord with field observations of the co-seismic ground ruptures along the fault but it's notably large compared to displacements estimated by seismicity, then

suggesting that most of the slip over the fault plane was aseismic". The problem is that, according to the Authors, the ground ruptures immediately after the main event seem to be in accord with the geodetic measurement: so, the displacement should be largely coseismic. ... Conversely, the Authors conclude that "only a part (from 5% to a maximum of 30%) of the stick-slip obtained by modeling is related to the co-seismic effects of the earthquakes recorded along the FF, suggesting that most of the slip over the fault must be aseismic." This inconsistency could be due to the scarcity of geodetic data, acquired in a limited number of campaigns from 1977 to 1980, and again only after the earthquakes in 1984, or to a mistake in data comparing. This issue could be easily addressed calculating the resulting S vector that should be larger than the measured ground rupture. Moreover, it is not clear which is the role of the similar ground rupture that affected the southeastern part of FF on occasion of the VIII EMS event of June 19 1984. Finally, in the Chapter Discussion and conclusions the last paragraph "These considerations again confirm the high level of seismic risk, in particular ground rupture hazard of the Fiandaca Fault and generally of the Timpe Fault System, for the several towns and villages located on these structures" should be deleted, since the authors have asserted before that most of the displacement is aseismic.

Other comments: 1) there are references from other research groups missing (see the attached pdf file); 2) the regional framework should be updated (see suggestions in the attached pdf file); 3) the formula of line 192 seems to be wrong, probably due to misprint; 4) computation of lines 296-297 should be extended; 5) the June 19 1984 event is rated as VIII EMS at line 145 and as VII EMS at line 308. 6) the straight dashed lines shown in fig. 4 are forced, being the eastern flank of Mt. Etna subject to episodic motion related to volcanic dynamics and gravitational motion;Other comments are listed in the attached pdf file.

Please also note the supplement to this comment:
http://www.nat-hazards-earth-syst-sci-discuss.net/nhess-2015-312/nhess-2015-312-

RC2-supplement.pdf
[Figure]

**Supplement:**

[revised manuscript text omitted]

---

## Author Comment (AC1) · 1 Apr 2016

*Replies to Comments from Maurizio Bonafede – Department of Physics and Astronomy, University of Bologna, Italy*

1) Several earthquakes struck the E flank of Mt Etna between 1980 and October 1984 (only one is mentioned in June 19th, 1984 but it is not accounted for); these may add transient components unresolved in the geodetic data between 1980 and 1984). Furthermore, as said already, deformation in a volcanic environment cannot be assumed as a steady state process, being related to episodic inflation/deflation episodes. For this reason, the straight dashed lines shown in figure 4 are meaningless.

**R: In order to answer the raised comment, we have added the following table that reports the parameters of 1980-1984 recorded earthquakes. Their locations have been also reported in the new figure 2B; the colored circles report the events close to the FF and STF and represent the most energetic events of the 1980-1984 periods.**

**Table 1. List of earthquakes recorded between May 1980 and October 1984 occurred in the investigated area (from Azzaro et al., 2000)**

|   | Date | Time | Longitude | Latitude | EMS | Md |
|---|------|------|-----------|----------|-----|----|
| 1 | 16/09/1980 | 0.104167 | 15.079 | 37.605 | VI | 2.9 |
| 2 | 26/11/1980 | 0.627778 | 15.118 | 37.723 | VI | 3.1 |
| 3 | 30/04/1981 | 0.522222 | 15.198 | 37.66 | VI | 3.5 |
| 4 | 13/09/1981 | 0.200694 | 15.161 | 37.647 | VI-VII | 3.3 |
| 5 | 06/07/1982 | 0.609028 | 15.104 | 37.698 | VI-VII | 3.8 |
| 6 | 20/07/1983 | 0.91875 | 15.096 | 37.603 | VII | 4.1 |
| 7 | 19/06/1984 | 0.638194 | 15.131 | 37.636 | VII | 3.4 |
| 8 | 19/10/1984 | 0.738194 | 15.103 | 37.694 | VII | 4.2 |
| 9 | 25/10/1984 | 0.049306 | 15.095 | 37.66 | VIII | 3.9 |

**We also agree with the reviewer comment about the assumption of linear trend and we have redrawn figure 4 removing the unrealistic dashed lines; moreover we have re-inverted the data without consider the trends, as required by the reviewer in a following comment.**

2) The formula at line 192 is wrong (hopefully it is only a misprint, otherwise the inversion procedure should be re-executed): the last term should read $e_{12} \sin 2d$ (not $e_{12} \sin^2 d$).

**R: We thank the reviewer for noticing this typo. We have corrected the formula**

3) It must be stated clearly that the deformation computed in this way is the "equivalent"

uniform deformation (as reported in figure 2.C) providing the same distance variations as the real non-uniform deformation, concentrated on the faults R: This comment is appropriate; we have included the suggested clarification in the 4.1 EDM Data paragraph. 4) Why are data at benchmarks 4 and 9 not taken into account? The number of free parameters (10) is so close to the number of independent data (13) that it is difficult to assess the reliability of the inversion.

**R: We don't take in account data at benchmarks 4 and 9 in order to exclude the MOF structure from inversion to reduce the number of free parameters. We are aware that the number of free parameters is close to the number of independent data, however we have now performed a goodness-of-fit test ($chi^2$ test) that assesses the inversion reliability for the given data.**

5) The more so, since data were arbitrarily (if I understand correctly, according to statement A above) corrected assuming a steady-state creep (dashed lines in figure 4) which is not supported by real data. Furthermore, some fault parameters (fault depth, length, dip) are fixed a priori. The data clearly show post-seismic creep and a major creep event is mentioned before the earthquakes (page 6). What would be the result of the inversion if the real data (1984 minus 1980) were considered?

**R: In order to overcome the correct issue raised by the reviewer we inverted the real data (1984 minus 1980). New results are reported in the following table and although slightly different form the previous ones they lead to similar considerations reported in the old manuscript. However in light of the new results we have reshaped the manuscript and made the new calculations for moments comparison and sensitivity parameters.**

|  | Fault 1 (FF) | | Fault 2 (STF) | |
|---|---|---|---|---|
| X (m, center top) | 509700 | Fixed | 512423 | fixed |
| Y (m, center top) | 4166660 | fixed | 4168595 | fixed |
| Depth (m, top) | 0 | fixed | 0 | fixed |
| Azimuth (°) | 140 | fixed | 142 | fixed |
| Dip (°) | 70 ± 0.0 | 70 - 89.9 | 70 ± 0.0 | 70 - 89.9 |
| Semi-Length (m) | 3500 ± 1200 | 2000 - 3500 | 2000 ± 0.0 | 2000 - 4000 |
| Width (m) | 3000 ± 0 | 1000 - 3000 | 2572 ± 1100 | 1000 - 3000 |
| Strike-s (cm) | 20.4 ± 1.6. | 0 - 100 (dextral) | 0.0 ± 0.0 | 0 - 100 (dextral) |
| Dip Slip (cm) | -12.7 ± 2.6 | -100 - 0 (normal) | -10.4 ± 5.7 | -100 - 0 (normal) |

*Minor points*

1. line 44: better write "... along the Timpe Fault System" instead of "... along the Fiandaca fault". **Ok we have changed it**

2. Lines 83-84: the previously unpublished data ... have been reviewed in the wake of new knowledge acquired in the last two decades (explain: what new knowledge? is it the fault parameters mentioned al line 206?), enabling insights into Etna's eastern flank ...

**Yes, we have added a paragraph explaining the knowledge acquired about the fault parameters and adding relevant references.**

3. The magnitude of the seismic events should be always stated when they are first mentioned.

**Yes, we have added the magnitude to the cited seismic events.**

4. line 297: I do not get the mentioned Mo values employing the magnitudes m=4.2 and m=3.9 mentioned in the text. Furthermore, these are duration magnitudes, not Richter magnitudes ML. In any case, it is clear that most of the fault displacement is aseismic.

**These comments are appropriates. We have revised calculations considering the local magnitude and we have rewritten the sentences as following:**

**An estimate of the seismic moment (Mo) release associated with the seismic events was obtained using the relation (Giampiccolo et al., 2007) for Etnean earthquakes:**

**Log(Mo) = (17.60 ± 0.37) + (1.12 ± 0.10)*ML**

**where ML is the local magnitude**

**Duration magnitude (MD) of 19 June and 25 October 1984 earthquakes were estimated in 3.4 and 3.9 (Table 1); we converted MD in local magnitude (obtaining 3.62 and 4.20 respectively) by using the Tuvè et al., (2015) relation:**

**ML = 1.164 (±0.011) *MD – 0.337 (±0.020)**

**Finally we obtained that Mo cannot be greater than = $1.2*10^{23}$ dyne-cm for the 25 October 1984 earthquake and $2.4*10^{22}$ dyne-cm for that on 19 June 1984.**

5. line 308: EMS VII is written here for the June '84 event while it is rated VIII at page 6.

**We have corrected in VII at page 6.**

6. explain acronym TDF at line 323; I cannot find it elsewhere; It's a misprint.

**We have changed it in STF**

7. the acronym MF is employed for both the Messina-Fiumefreddo line and for Moscarello fault: consider revising.

**We thank the reviewer for revealing this ambiguity. We have changed in MOF the acronym of Moscarello fault**

8. Figure 2: consider reporting in the caption the acronyms of the faults. Eliminate topographic level lines from panel B. Write "instrumental epicenter" (instead of "analytic

location") and "macroseismic epicenter" otherwise (if I understand correctly).

**We have modified figure 2 considering these suggestions.**

9. Figure 3 is unnecessary: consider deleting, leaving the web link in the text.

**We have removed figure 3**

*References*

- Azzaro, R., M.S. Barbano, B. Antichi and R. Rigano 2000. Macroseismic catalogue of Mt. Etna earthquakes from 1832 to 1998, Acta Vulcanologica, 12 (1-2), 3-36.

- Giampiccolo, E., S. D'Amico, D. Patanè, and S. Gresta, 2007, Attenuation and source parameters of shallow microearthquakes at Mt. Etna Volcano, Italy, Bull. Seismol. Soc. Am., 97, 184–197, doi:10.1785/0120050252.

- Tuvè T., D'Amico, S., Giampiccolo E. (2015). A new MD-ML relationship for Mt. Etna earthquakes (Italy). Annals of Geophysics, 58, 6, doi:10.4401/ag-6830S0657.
* * *
[Figure]

**Fig. 1.**

[Figure]

**Fig. 2.**

---

## Author Comment (AC3) · 1 Apr 2016

*Replies to Comments from Reviewer #2*

1) For example it is necessary to frame the aseismic slip in process of gravitational sliding involvig the eastern flank of Mt. Etna. In particular , the Authors assert that "in the May 1980-October 1984 period, the Fiandaca Fault was affected by a strike slip and normal dip slip of about 27 and 23 cm. This result is in fairly good accord with field observations of the co-seismic ground ruptures along the fault but it's notably large compared to displacements estimated by seismicity, then suggesting that most of the slip over the fault plane was aseismic". The problem is that, according to the Authors, the ground ruptures immediately after the main event seem to be in accord

with the geodetic measurement: so, the displacement should be largely coseismic. . .. Conversely, the Authors conclude that "only a part (from 5% to a maximum of 30%) of the stick-slip obtained by modeling is related to the co-seismic effects of the earthquakes recorded along the FF, suggesting that most of the slip over the fault must be aseismic." This inconsistency could be due to the scarcity of geodetic data, acquired in a limited number of campaigns from 1977 to 1980, and again only after the earthquakes in 1984, or to a mistake in data comparing. This issue could be easily addressed calculating the resulting S vector that should be larger than the measured ground rupture.

**R: We thank the reviewer for this comment. On lines 327-330 we wrote: "This result is in some ways comparable with field observations that detected a co-seismic ground rupture of the northwestern and southeastern sectors of the fault of up to 20 cm (Azzaro, 1999 and reference therein), while a discrepancy between the seismic and geodetic moment is present." We have made a mistake in defining the ground rupture (measured in the days after the earthquakes) as "co-seismic ground rupture" cause the ground ruptures linked to shallow earthquakes on Mt. Etna are the sum of coseismic and aseismic movements (e.g. Obrizzo et al., 2001). In this sense, we have rewritten the sentence as following: "This result is in some ways comparable with field observations that detected ground ruptures (co-seismic and aseismic) of the northwestern and southeastern sectors of the fault up to 20 cm (Azzaro, 1999 and reference therein). The discrepancy between the seismic and geodetic moment allows us to quantify the amount of aseismic deformation."**

2) Moreover, it is not clear which is the role of the similar ground rupture that affected the southeastern part of FF on occasion of the VII EMS event of June 19 1984.

**R: This is an interesting comment: we think that the ground rupture that affected the southeastern part of FF (on occasion of the VII EMS event of June 19 1984) represents the first phase of a process that involved all the FF ending on 25**

**October 1984. Indeed, a northwards "migration" of the rupture sequence was suggested by Azzaro (1999) and our model is in accordance with this hypothesis showing that the entire FF was activated (even if data do not allow us to discern the time-space sequence). We have added these considerations in the discussion/conclusions.**

3) Finally, in the Chapter Discussion and conclusions the last paragraph "These considerations again confirm the high level of seismic risk, in particular ground rupture hazard of the Fiandaca Fault and generally of the Timpe Fault System, for the several towns and villages located on these structures" should be deleted, since the authors have asserted before that most of the displacement is aseismic.

**R: We thank the reviewer for this comment. We made a mistake in using the term "seismic risk". We have changed it in "geological hazard" embracing both the hazard due to seismic shaking effects and the one due to ground rupture effects.**

Other comments:

1) there are references from other research groups missing (see the attached pdf file);

**All the suggested references have been added to text.**

2) the regional framework should be updated (see suggestions in the attached pdf file);

**We have accepted all the suggestions.**

3) the formula of line 192 seems to be wrong, probably due to misprint;

**It's a misprint we have corrected the formula**

4) computation of lines 296-297 should be extended;

**We have extended the computation of lines 296-297 also taking in account the reviewer 1 comments (see reviewer 1 reply).**

5) the June 19 1984 event is rated as VIII EMS at line 145 and as VII EMS at line 308.

**We have corrected in VII EMS at line 145.**

6) the straight dashed lines shown in fig. 4 are forced, being the eastern flank of Mt. Etna subject to episodic motion related to volcanic dynamics and gravitational motion;

**We have redrawn figure 4 removing the dashed lines**

Other comments are listed in the attached pdf file

**All the comments on the attached pdf file have been considered and accepted.**

*References*

- Azzaro, R. 1999. Earthquake Surface Faulting At Mount Etna Volcano (Sicily) And Implications For Active Tectonics, Journal of Geodynamics, 28 (2-3), 193-213.

- Obrizzo, F., Pingue, F., Troise, C., De Natale, G., 2001. Coseismic displacements and creeping along Pernicana fault (Mt. Etna) in the last seventeen years: a detailed study of a structure on a volcano. Journal of Volcanology and Geothermal Research 9, 109–131.

[Figure]

Fig. 1.